# OBELISK – One Kernel to Solve Nearly Everything: Unified 3D Binary Convolutions for Image Analysis

Mattias P. Heinrich[1], Ozan Oktay[2], and Nassim Bouteldja[1]

[1]Institute of Medical Informatics, University of Lübeck, DE
[2]Babylon Health, London, UK
heinrich@imi.uni-luebeck.de

## Abstract

Deep networks have set the state-of-the-art in most image analysis tasks by replacing handcrafted features with learned convolution filters within end-to-end trainable architectures. Still, the specifications of a convolutional network are subject to much manual design – the shape and size of the receptive field for convolutional operations is a very sensitive part that has to be tuned for different image analysis applications. 3D fully-convolutional multi-scale architectures with skip-connection that excel at semantic segmentation and landmark localisation have huge memory requirements and rely on large annotated datasets - an important limitation for wider adaptation in medical image analysis.

We propose a novel and effective method based on a single trainable 3D convolution kernel that addresses these issues and enables high quality results with a compact four-layer architecture and without sensitive hyperparameters for convolutions and architectural design. Instead of a manual choice of filter size, dilation of weights, and number of scales, our *one binary extremely large and inflecting sparse kernel* (OBELISK) automatically learns filter offsets in a differentiable continuous space together with weight coefficients. Geometric data augmentation can be directly incorporated into the training by simple coordinate transforms. This powerful new architecture has less than 130'000 parameters, can be trained in few minutes with only 700 MBytes of memory and achieves an increase of Dice overlap of $+5.5\%$ compared to the U-Net for CT multi-organ segmentation.

## 1 Introduction

Given its undeniably superior performance, deep convolutional neural networks (DCNN) have replaced most existing approaches for image classification, dense segmentation, pre-processing, object or landmark localisation. Early theoretical work proved that a wide enough two layer neural network can approximate any highly complex nonlinear transfer function. Yet, despite difficulties to explain an underlying reasoning the following *common belief or general wisdom* has evolved:

1) "more convolutional layers always lead to better results" e.g. [6],

2) "small convolutional filters are preferable to larger ones" e.g. [24], and

3) "multi-scale or progressive dilated receptive field are necessary for dense prediction" e.g. [17, 27].

These beliefs have rarely been questioned, mainly because conventional convolutional filter representations are too rigid in their design to approach the ideal of a compact, shallow and low-parametric network with sufficient generalisation quality. In this work, we present a new concept that should help to overcome this *mind barrier* and **open new possibilities of CNN architectures that are much easier to design, train and deploy** - with great potential impact on medical image analysis.

1st Conference on Medical Imaging with Deep Learning (MIDL 2018), Amsterdam, The Netherlands.

Commonly used convolutional kernels have a user-defined layout of sampling locations that is usually restricted to a regular $3 \times 3$ (or $3 \times 3 \times 3$) grid and only filter coefficients are automatically learned. In order to capture both local and regional context, several of these small $3 \times 3$ kernels are concatenated and supplemented by strided pooling operations that reduce resolution. To restore local detail in dense prediction tasks, i.e. semantic segmentation or localisation, upsampling or fractionally strided layers are employed that increase resolution. We propose a radically different solution. Our concept is inspired by recent work on differentiable interpolation in CNN architectures for spatial transformers [11], deformable convolutions [3] and sparse binary convolutional kernels [9, 13, 7]. We propose a **new binary kernel** in which both the **spatial offsets and coefficients** are learned in a continuous, differentiable space. To deal with the challenges that a neural network has to face when encountering features and object relations of different scales, we choose a sufficiently large number of filter elements that can replace manually designed multi-scale architectures (e.g. [18, 21]) using only a single large kernel. By learning the spatial filter offsets, the network can decide on its own how sparse and how large the receptive field has to be for a given task. In our experiments, we can see that even with a very narrow random initialisation that is in the range of few pixels, our kernel quickly enlarges drastically in size to capture all relevant contextual information.

We further note that current fully convolutional architectures lose some of the important benefits of stochastic gradient descent optimisation. Because encoder-decoder architectures are only computationally efficient when trained with large image patches (or the whole image) in parallel, the variability in each batch is severely reduced. It was claimed empirically in [17] that using only very few distinct samples per batch and averaging the gradients across 10 thousands of pixels within the same image does not hurt convergence. However, in the medical context, where usually only dozens of labelled 3D scans are available for training, this assumption may no longer hold. Training is substantially slower and may require data augmentation when using these large nearly uniform batches of pixels belonging to the same image. Subsequently, we propose a second simplification and remove any subsequent spatial convolutions following the *one binary extremely large and inflecting sparse kernel* (OBELISK). Instead, we simply use four $1 \times 1$ convolutions [16] (with batch norm and ReLU) that form a multi-layer perceptron shared among all locations for dense pixelwise predictions. That way, we can benefit from more diverse sampling during stochastic gradient descent optimisation and reach convergence much faster. In addition, it also directly offers a simple solution to the important problem of class-imbalance in medical image segmentation. By performing simple online hard example mining based on the current training loss, we can achieve very accurate results in terms of Dice overlap without ever specifying it directly as cost function (Dice loss) [18], which can cause problems due to its poor differentiability.

## 2   One Binary Extremely Large and Inflecting Sparse Kernel

The key to our novel convolutional architecture is a large layer that learns both filter offsets and weights automatically from the given data. By inflecting (spatially adapting) the local offsets directly in a data-driven way, the manual design of convolutional architectures can be omitted and the best setting for the given problem can be automatically learned. While this is inspired by spatial transformer networks [11] and deformable convolutions [3], our approach is more general in that the learned kernel is not dependent on a separately estimated class or object geometry prediction. Differently, we aim to learn a generic kernel that is applicable without spatial transformation and can replace multiple small filter kernels at different scales (see Fig. 1). Our learned kernel can be very sparse and substantially increase the receptive field and spatial context aggregation, an important aspect for medical image analysis as shown in previous works [2, 9]. In turn, it enables us to train a network with only 130'000 free parameters that achieves remarkably accurate predictions, uses very little memory (<700 MBytes for dense 3D segmentation) and is very fast to train.

Consider a classical 2D convolution operation for a kernel with 25 elements (forming a $5 \times 5$ filter) and dilation factor of $d$ [26, 27]. The spatial filter offsets are statically defined as $(s_x, s_y) = \{-2d, d, 0, +d, +2d\}^2 \in \mathbb{Z}^{5 \times 5}$. Let $I(x, y)$ be the value of an input at location $(x, y)$ and $W \in \mathbb{R}^{5 \times 5}$ the continuous valued and trainable filter coefficients. The output $F(x, y)$ can be calculated as:

$$F(x,y) = \sum_i \sum_j W(i,j) \cdot I(x + s_x(i,j), y + s_y(i,j)) \tag{1}$$

Since, both the pointwise multiplication and the sum operation are differentiable, we can easily find the derivate of a convolution operation with respect to the weights $W$ and the input $I$. Let us now

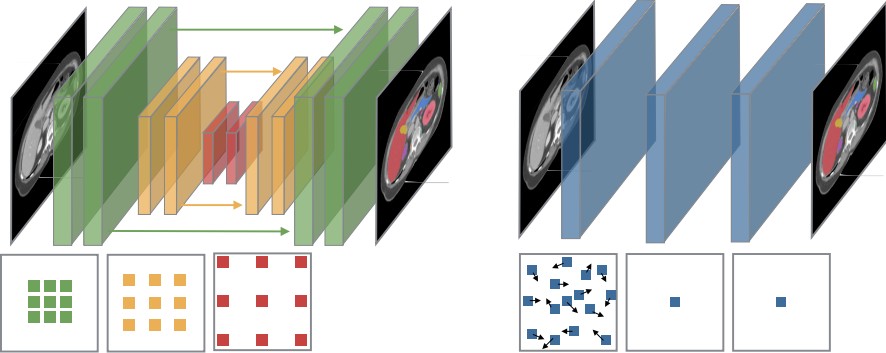

Figure 1: Visual comparison of fully-convolutional multi-scale architectures (left) with our new OBELISK method (right, blue). Instead of having multiple small convolution layers that have to be carefully designed in scale (dilation) and size, we propose to use one extremely large sparse kernel followed by only channel-wise $1 \times 1$ convolutions. Thereby, the output of each voxel can be computed independently and much fewer parameters are required, while all parameters are shared across sampling locations (translational invariance). The spatial offsets of this kernel are continuously defined and end-to-end trainable.

consider the continuous valued filter offsets $S_x \in \mathbb{R}^{5 \times 5}$ and $S_y \in \mathbb{R}^{5 \times 5}$. To obtain the convolution output for inputs on a discrete grid, we need to perform bilinear interpolation:

$$F(x,y) = \sum_i \sum_j W(i,j) \cdot (w_1 I(\lfloor x + S_x(i,j) \rfloor, \lfloor y + S_y(i,j) \rfloor) + w_2 I(\lceil x + S_x(i,j) \rceil,$$
$$\lfloor y + S_y(i,j) \rfloor) + w_3 I(\lfloor x + S_x(i,j) \rfloor, \lceil y + S_y(i,j) \rceil) + w_4 I(\lceil x + S_x(i,j) \rceil, \lceil y + S_y(i,j) \rceil))$$
$$(2)$$

with the following bilinear coefficients $w_1, \ldots, w_4$:

$$\begin{aligned}
w_1 &= (\lfloor x + S_x(i,j) + 1 \rfloor - (x + S_x(i,j))) \, (\lfloor y + S_y(i,j) + 1 \rfloor - (y + S_y(i,j))) \\
w_2 &= (x + S_x(i,j) - \lfloor x + S_x(i,j) \rfloor) \, (\lfloor y + S_y(i,j) + 1 \rfloor - (y + S_y(i,j))) \\
w_3 &= (\lfloor x + S_x(i,j) + 1 \rfloor - (x + S_x(i,j))) \, (y + S_y(i,j) - \lfloor y + S_y(i,j) \rfloor) \\
w_4 &= (x + S_x(i,j) - \lfloor x + S_x(i,j) \rfloor) \, (y + S_y(i,j) - \lfloor y + S_y(i,j) \rfloor)
\end{aligned} \quad (3)$$

Again all operations (multiplications, min/max for floor/ceil, and additions) are differentiable. We can therefore obtain the derivatives with respect to the filter coefficients $W$, their spatial offsets $S_x, S_y$ and the input if necessary. We employ our approach mainly for 3D applications and the extensions of Equations 2 and 3 to trilinear interpolation using 8 positions and interpolation weights is straightforward.

We refer to learning one spatial 3D offset per filter coefficient as *unary variant* and propose a *binary* extension that learns two offsets for each convolution element. In this case the two interpolated values from the preceding layer (the image input) are first subtracted and the outcome is multiplied with their joint filter weight.

## 2.1 Implementation details

The spatial offsets $S_x, S_y, S_z$ are initialised with normally distributed random numbers and zero mean just as their coefficients are. Fig. 2 (left) shows an example of the spatial distribution of filter offsets. Similar to previous work (e.g. [27]), we empirically found that for very large receptive fields a sufficient throughput of local information is necessary. Therefore, smaller values for the standard deviation $\sigma$ of the normal distribution are preferable. We used $\sigma = 0.05$ – the image coordinates range from -1 to +1, but $\sigma = 0.02$ to $\sigma = 0.1$ gave almost identical results. The network will automatically learn to increase the receptive field if necessary to as much as half of the image domain.

**Network architectures:** Since, we rely on only this single spatial convolution filter it requires as many as 1024 spatial 3D offset elements. When employing our binary variant that pairs two offsets for

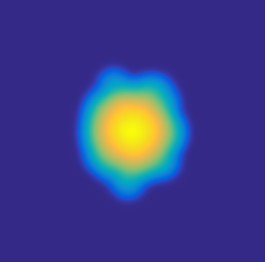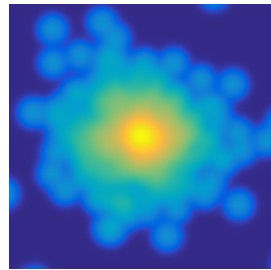

Figure 2: Example of spatial distribution of filter offsets ($S_x$, $S_y$ are shown, $S_z$ omitted) – shown with logarithmic colormap – at the start of the training of the OBELISK layer (left) and after optimisation for 50 epochs (right). The size of the extent of the figures corresponds to half of the image domain. It is evident that our data-driven approach yielded a larger receptive field, but still contains many filter coefficients close to the centre.

the next layer and subtracts their values, this number is doubled. While this may seem excessive, we note, that a standard $3 \times 3 \times 3$ kernel for 64 channels has already more input features. The OBELISK layer is followed by a small number of $1 \times 1$ convolutions to learn complex spatial patterns from the data for dense prediction tasks. That means it is necessary to learn $6 \times 1024$ spatial offsets and 1024 coefficients for each subsequent feature channel. To reduce the number of weights in the first layer, we use 4 channel groups and 256 feature channels (yielding 71'680 trainable parameters for OBELISK). Afterwards, we explore two different network architectures. First, a simple multi-layer perceptron that employs two layers with 256 and 128 feature channels together with batch normalisation and standard ReLU activation. Second, a slightly more complex $1 \times 1$-DenseNet with an initial channel reduction to 128 features, followed by four layers with growth rate of 32 and feature concatenation (re-use) [10]. This results in a prediction layer with 256 input features. In total both networks are extremely light-weight with less than 130'000 parameters.

**Training locations:** Another key advantage is that this new concept does not rely on fully-convolutional training anymore. In fact the image locations for which we draw labels and calculate the large convolution kernel can be on arbitrary continuous-valued coordinates. This enables us to use smaller mini-batches (e.g. 128 samples) that are randomly selected across different images and locations and are thus more diverse than the large image batches used in FCNs. This can also be exploited for adaptive sampling for landmark localisation or sparse deformable registration (see below). It furthermore gives the opportunity for online hard example mining (OHEM) [23]. A technique that only selects a quantile of largest training errors for backpropagation of the loss and thereby increases the difficulty-awareness for semantic segmentation [15]. Our implementation[1] relies on the excellent differentiable `grid_sample` function available for up to 5D tensors (three spatial dimensions) in pytorch. But we will also make available our own naive implementation (that is slower) for researchers interested in porting the functionality to other toolboxes. Integrating geometric data augmentation directly within our approach is also straightforward and very elegant (as opposed to transforming the whole 3D image for conventional FCNs). We simply perform a (batched) matrix multiplication for the sampling offsets $S_x, S_y, S_z$ using random $3 \times 3$ matrices (with Gaussian standard deviation of 0.2 plus an identity transform). Similar to the idea proposed in [1] (as shape-indexed features) this has the same effect as transforming the whole input scan, but enables increased variability by applying a different affine matrix for each sample location.

**Limitations and possible extensions:** Our current implementation and experimental setup has a number of limitations that could be addressed in future work. While training times are decreased compared to U-Net, the inference for unseen data may take longer (20 sec. vs 1.5 sec. using a dense grid with stride 2), which is mainly due to the less predictable and therefore slower memory access in our OBELISK kernel. This drawback could be alleviated by progressively refining the sampling grid during inference (similar to an agent based landmark search c.f. [4]). We performed some initial experiments for landmark localisation of the adrenal glands – a tiny anatomical structure that consists of only $\approx 1'000$ voxels (see also [20] for related work using CNNs). We used 6 iterations of progressive refinement (using a single model trained for multi-label segmentation), where each subsequent coordinate sampling depends on locations with high anatomy probability. Compared

---

[1]source code available at `https://github.com/mattiaspaul/OBELISK`

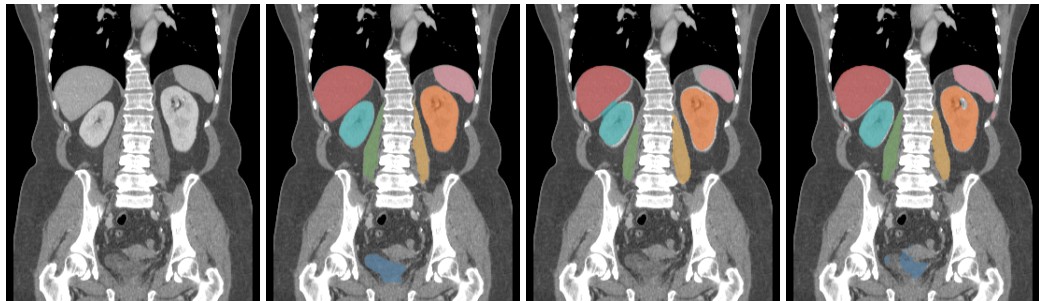

Figure 3: Comparison of segmentation overlays for seven anatomical structures: ■ liver, ■ spleen, ■ bladder, ■ left kidney, ■ right kidney, ■ left psoas major muscle (pmm) and ■ right pmm. From left to right: coronal plane of original CT scan, ground truth segmentation, U-Net multiscale architecture and proposed OBELISK network (without data augmentation). A much better segmentation of the unary bladder and more detailed delineation of psoas muscles as well as a clearer differentiation between liver and kidney are visible for our approach.

to dense sampling, the median localisation accuracy is only slightly reduced from 5.0 to 5.9 voxels (tested for 7 CT scans), while using 95% less samples.

We have not yet experimented with multi-layer OBELISK convolutions, which is obviously of interest. On the one hand, this would require sampling on regular grid (for subsequent convolutions), reduce variability within mini-batches and increase memory demand. On the other hand, it would enable deeper networks that could gain from more non-linearities while being able to cover large enough receptive field with smaller kernels through concatenation. An alternative way of increasing depth without losing the benefits of a single layer convolution kernel are binary tree [28] or densely connected [10] deep architectures for $1 \times 1$ convolutions. That means that starting from many potential kernel offsets, multiple pathways through the network could be used to learn the optimal depth for sharing features across different contextual scales.

## 3   Experiments and Results

We performed initial experiments with leave-one-out cross validation on 10 contrast-enhanced 3D CT scans of the VISCERAL3 training dataset [12] and segmented the following seven anatomical structures: ■ liver, ■ spleen, ■ bladder, ■ left kidney, ■ right kidney, ■ left psoas major muscle (pmm) and ■ right pmm (see Fig. 3). For pre-processing the images were resampled to isotropic voxel sizes of 1.5 mm$^3$. This is in general more important for the compared U-Net architecture, since the OBELISK layer can learn to deal with anisotropic spacings (as long as it is consistent). The volumes were cropped to dimensions of $312 \times 230 \times 320$ voxels, without using any guidance information. Note that such a rough cropping poses a much harder challenge than two-stage approaches [22], regional CNNs [14] or accurate manual bounding box selection [5]. To ease the learning for the U-Net (and reduce its substantial memory demand to 2.5 GByte) the images were smoothed with average pooling and downsampled by a factor of two. For the OBELISK approach a slightly stronger smoothing ($\sigma = 1.5$) was applied but no downsampling, since the memory requirement (of roughly 700 MByte) is independent of image dimensions. All network layers are trained with the same learning rate of 0.002 (using Adam). For our proposed architecture, we used a batch size of 192, and 64 iterations (of random mini-batches) per epoch. Online hard example mining [23] was used with quantiles of 75% or 50% (and respectively increased batch sizes, see Table 1 for details).

The U-Net architecture used for comparison consists of five resolution levels with in total fourteen $3 \times 3 \times 3$ convolutional layers and 8–128 feature channels (the largest input in the expanding part has 192 input channels and carries therefore alone already more parameters than the entire OBELISK network). Extensive efforts have been made to enable the best results for the U-Net, we found in particular that a leaky ReLU (with negative slope = 0.1) helped a lot and we increased the number of training epochs from 50 to 100. As proposed for 3D U-Nets in [19], we used strided convolutions in the contracting part and double the number of filters thereafter. We experimented with a class-weighted cross-entropy loss and the Dice loss [18] but this led to lower performance.

Table 1: The quantitative evaluation of a leave-one-out cross-validation for 10 scans of the VIS-CERAL3 training database demonstrates the advantages in terms of Dice accuracy (**+5.5%**) and reduction of parameters (ten-fold) of our proposed OBELISK approach compared to the state-of-the-art multiscale U-Net. The memory requirement for our proposed OBELISK network is $\approx$700 MByte, while the U-Net requires more than 2'500 MByte. **++** defines online augmentation using affine transformations of OBELISK offsets without any additional image manipulation. **OHEM** stands for online hard example mining, which simply back-propagates only the top $^1/_2$ or $^1/_4$ fraction of individual loss terms during training.

| Method | Parameters | Batch-Size | Dice-Score |
|---|---|---|---|
| OBELISK + $1 \times 1$-Dense | 129'536 | 192 | 73.68% |
| OBELISK + $1 \times 1$-Dense | 129'536 | 384 (OHEM $^1/_2$) | 75.85% |
| OBELISK + MLP | 122'368 | 384 (OHEM $^1/_2$) | 75.62% |
| Unary OBELISK + Dense | 126'464 | 384 (OHEM $^1/_2$) | 72.32% |
| Rand. Offsets + $1 \times 1$-Dense | 123'392 | 384 (OHEM $^1/_2$) | 71.67% |
| **OBELISK + $1 \times 1$-Dense** | 129'536 | 768 (OHEM $^1/_4$) | **76.68%** |
| OBELISK + $1 \times 1$-Dense ++ | 129'536 | 768 (OHEM $^1/_4$) | 80.61% |
| multiscale U-Net | 1'250'000 | 6000 | 71.12% |

Our experimental comparison showed a significantly better performance of our much simpler architecture with OBELISK compared to the 3D U-Net ($p = 0.015$ using ranksum test) with average Dice scores of 76.7% vs. 71.1% (see Table 1). Interestingly, the differences between DenseNet and classical MLP classifiers (after OBELISK) are very small and a clear advantage of learning offsets (compared to simply using random initialisation with a tuned standard deviation) is obvious. The *unary* variant refers to using only one 3D offset per filter coefficient, as compared to our proposed binary pairs, performs substantially worse, probably due to stronger constraints that are obtained when tying two offsets to one coefficient. The learning curves and box-plots shown in Fig. 4 demonstrate the hugely positive effect in training speed when using OBELISK and a further advantage of online hard example mining and data augmentation. The individual Dice scores per label reveal advantages for the kidneys and, when using affine augmentation, also for bladder and spleen compared to the multiscale U-Net.

Further substantial improvements in accuracy are easily possible by using a larger corpus of training data (e.g. the one made recently available by [5]), the use of post-processing such as edge-preserving smoothing [8] and more aggressive data augmentation. However, we already achieve comparable accuracy with respect to other published work on the VISCERAL3 dataset (despite using only 9

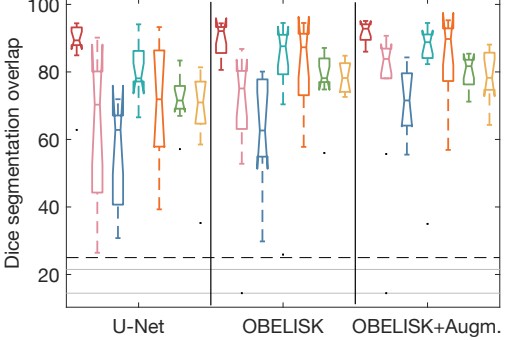 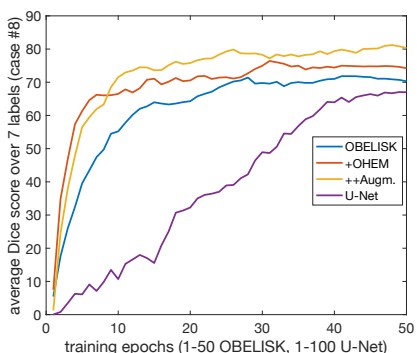

Figure 4: Dice overlaps for leave-one-out validation drawn as box-plots on the left for each anatomical label (■ liver, ■ spleen, ■ bladder, ■ left kidney, ■ right kidney, ■ left psoas major muscle (pmm) and ■ right pmm) shown improved accuracy for the OBELISK concept for kidneys and spleen and the benefits of online data augmentation for psoas muscles. The better ability to randomly shuffle training examples using the OBELISK architecture yields much faster training (example validation curve is shown on the right), which can further be improved by online hard example mining (OHEM).

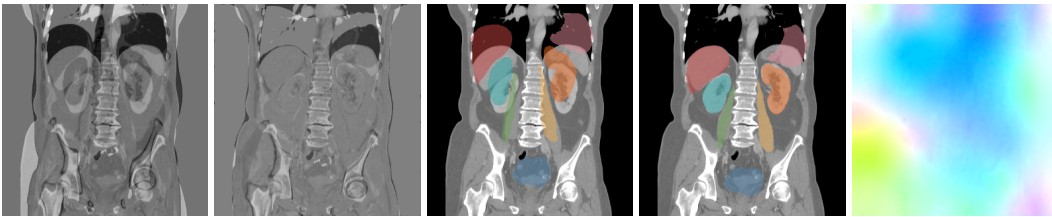

Figure 5: Proof-of-concept for deformable registration within the same network architecture. This time optimising the sampling locations of the moving image to minimise their feature difference in comparison to those of the fixed image (learned for segmentation task). From left to right: Difference image before, and after registration; propagated segmentation labels without and with registration; and estimated deformation field using RGB flow-colours. See Fig. 3 for colour definition for anatomical labels.

training images). Considering only liver, kidneys, spleen and psoas muscles (to enable comparison to [25]), our approach reaches an average Dice of 82.2%, keypoint-transfer [25] yielded 78%, multi-atlas label fusion (according to [25]) 70%, while the best performance of MALF by Kechichian et al. on the no-longer available test set [12] was 88% (in our experience, the test set has slightly easier scans).

## 3.1 Deformable registration using OBELISK features

To demonstrate the ability to transfer knowledge of the proposed binary sparse convolution kernels to capture an extremely large context, we performed deformable (2D) registration based on the learned features (trained for a segmentation task). For the fixed image, a forward pass up to the second last layer yields 128-dimensional feature vectors for all spatial sampling coordinates. The only change to our architecture is that we now freeze the optimisation of filter offsets and convolutional weights, but minimise an L1 loss term (on the difference of feature vectors between fixed and moving image) by changing the sampling coordinates of the moving image. In order to obtain a smooth deformation field, an additional penalty that penalises the difference between the estimated deformations and a B-spline filtered version of them, is introduced. The visual outcome of this proof-of-concept experiment is shown in Fig. 5.

## 4 Discussion and Conclusions

We have presented a novel convolutional architecture that enables accurate dense predictions, demonstrated on the example of 3D multi-organ segmentation, using one binary extremely large and inflecting sparse kernel (OBELISK). It unifies previous multi-resolution, cascaded dilatation and deformable convolution approaches into a simple framework that consists of an OBELISK layer and subsequently only uses $1 \times 1$ convolutions. Our initial experimental results, which outperform 3D U-Nets by 5.5% Dice overlap for multi-organ segmentation while reducing the number of parameters by 90%, indicate that the three common requirements for CNN-based segmentation (deep networks, small kernels, multi-scale or dilation architecture) are not always necessarily the best choice.

Our concept goes beyond a fixed pixel grid for filter coefficients in convolutions and also beyond a regular sampling grid for extracting information from images. This provides at least three important advantages to conventional fully-convolutional architectures. First, it can deal more naturally with anisotropic highdimensional data, which is important in 3D medical imaging and also for temporal signals (2D+t). Second, the spatial locations for predictions can be adaptively sampled also for off-grid positions, which enables an increased focus on hard examples in the data, more variability in mini-batches and thus faster training and substantially reduced memory requirements. Third, features can be extracted with true translational invariance (in contrast to U-Nets) and for arbitrary positions, which can be beneficial for further tasks such as deformable registration (for which we show a proof-of-concept).

We further note that the receptive field in a U-Net is directly coupled to the number of scales. Thus the resolution of the usable grid with the most contextual information (coarsest scale) contains only every 512th voxel (example for four-level U-Net with strides of $2^3 = 8$) and yields a receptive field of only $61 \times 61 \times 61$ voxels. The OBELISK concept de-couples these aspects and might therefore lead

to better fine-grained recognition of small anatomical structures. As extension to previous deformable convolutions, we found that sampling two offsets per filter element and thereby introducing an implicit weight sharing (with opposite sign) seems to stabilise the learning of the spatial kernel layout (and improves the average Dice by 4%). We also derive a straightforward implementation for online affine augmentation that can independently transform the neighbourhood of every feature location.

In future work, we will explore larger public datasets that would further increase performance and allow for a more elaborate validation. The use of post-processing and/or multi-layer convolutions could be integrated and the concept can be transferred to a number of related tasks, such as landmark localisation and weakly-supervised deformable registration. The use of regularisation strategies (dropout, weight decay) and deeper supervision should also be explored. The use of online hard example mining is not restricted to our approach and has great potential to also improve the convergence of U-Nets and other FCN architectures used in medical imaging.

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
