# OpenReview forum: "OBELISK - One Kernel to Solve Nearly Everything: Unified 3D Binary Convolutions for Image Analysis"
_MIDL.amsterdam/2018/Conference — MIDL 2018 Oral_

### Review · AnonReviewer3 · 2018-05-07
**Interesting idea, incomplete evaluation**

**Rating:** 3
**Confidence:** 2

**Review:**

The authors present a convolution kernel that learns not only the convolution weights, but also element-wise pixel offsets in continuous space. This theoretically allows the network to learn contextual information at each level of the network automatically, as opposed to current approaches of manually specifying kernel dilation on a fixed rectangular grid.

This is an interesting idea. The theory is well explained; Figures 1 and 2 add significantly to understanding of the overall approach.

The implementation details should be clearer, specifically in Section 2.1 on "Network architectures"; it is hard to follow and see where all the numbers are coming from. The results portion of the paper also seems incomplete. For instance, this paper seems like it should include networks with multiple OBELISK-based layers to fully demonstrate the proof-of-concept.

The deformable registration example appears to be a throw-in and does not really add to this particular paper. If the approach works well and can be reasonably demonstrated, then it would be worthy of a separate publication.

Overall, the general approach will be interesting to a wide audience and the overall concept could mature into something useful. However, the concepts and work presented in this paper is clearly in its infancy.

**Special Issue:**

No

---

### Review · AnonReviewer1 · 2018-05-09
**A well-written and thorough paper addressing relevant issues with a very novel algorithm.**

**Rating:** 5
**Confidence:** 2

**Review:**

The authors describe an alternative to the conventional deep convolutional network whereby a single kernel is used to automatically learn concepts such as scale and filter size.  This method, as well as removing elements of architectural decision-making, additionally has much lower memory requirements than its competitors.  The paper is very well written and describes an extremely novel and very relevant method which potentially addresses a number of issues in deep learning for medical imaging.  The experiments are well explained and thorough and are compared with a state-of-the-art method for segmentation (U-Net).  Future potential and limitations are addressed adequately.  I consider this a very promising research direction and have no hesitance in recommending this paper for acceptance.  I have only very few and minor comments or criticisms:
 - The concept of spatial offsets should be described in a brief sentence since these are may not be familiar to the user of conventional CNNs.
 - There is a great deal of detail packed into this paper, which is a positive thing, but at times it seems that the space constraints put the authors at a disadvantage in that they cannot fully explain some elements of the architecture (e.g. the pairing and subtracting offsets in subsequent layer).  The experiment with registration at the end of the paper is interesting, but very brief and not quantitatively evaluated.  It would be my preference to remove this experiment in favour of more detailed descriptions/figures elsewhere in the paper.

**Special Issue:**

Definitely

---

### Review · AnonReviewer2 · 2018-05-17

**Rating:** 3
**Confidence:** 3

**Review:**



**Special Issue:**

No

---

### Review · AnonReviewer4 · 2018-05-21

**Rating:** 5
**Confidence:** 2

**Review:**

This paper is interesting because learning filter offsets directly is still relatively unexplored. The authors follow ref 3, is this the only paper that tried to do this?

The binary part, the B in OBELISK, is not well explained. The paper says: "The unary variant refers to using only one 3D offset per filter coefficient, as compared to our proposed binary pairs, performs substantially worse, probably due to stronger constraints that are obtained when tying two offsets to one coefficient.

The part "We additionally found it beneficial to pair two offsets for the next layer and subtracting their values, which further doubles this number." also hard to follow.

I agree with the other reviewers that the deformation experiment is a bit preliminary. It would be better to perform more extensive evaluations just for segmentation.

There is one typo OBELSIK.

**Special Issue:**

Yes

---

### Comment · ~Bram_van_Ginneken1 · 2018-05-18
**Selection for longlist for special issue Medical Image Analysis**

Dear authors,

Congratulations on your acceptance to MIDL! We have selected your paper on the longlist for the Medical Image Analysis Special Issue. Please read this page:
https://midl.amsterdam/special-issue-in-medical-image-analysis/
Please answer the three questions that are listed on that page about your interest in submitting to the special issue, potential overlap with other publications, and related publications.

You can post your answer here directly below on openreview.net, or mail me directly at bram.vanginneken@radboudumc.nl.

Best regards, Bram

---

### Decision · Program_Chairs · 2018-05-15
**Paper20 Acceptance Decision**

Oral